# Comparative analysis of mental and physical fatigue on motor coordination, visual search patterns, perceived effort, and performance outcomes in closed-loop aiming task

**Maryam Khalaji**[1]*, **Parvaneh Shamsipour Dehkordi**[1]*, **Fatemehsadat Mousavian**[1], **Saeed Alboghebeish**[2]

**1** Department of Motor Behavior, Faculty of Sport Sciences, Alzahra University, Tehran, Iran,
**2** Department of Sport Sciences and Health, Shahid Beheshti University, Tehran, Iran

* m.khalaji@alzahra.ac.ir (MK); p.shamsipour@alzahra.ac.ir (PSD)

## Abstracts

This study has investigated the effects of both mental and physical fatigue on visual search patterns, motor coordination, perceived exertion, and performance accuracy while executing the golf swing. Thirty female recreational golfers with no professional experience in golf, aged $21.70 \pm 2.21$, underwent both mental and physical fatigue protocols in two separate conditions. In addition to the radial error, swing accuracy, Quiet Eye Duration (QED), and fixation number were recorded before and after fatigue. The results showed that mental fatigue significantly decreased QE duration (Mean difference $= -138.75$ ms, $p = .0009$) and fixation duration (Mean difference $= 67.50$ ms, $p = .001$), indicating a detrimental effect on sustained attention. Physical fatigue led to more fixations (Mean difference $= 1.18$, $p = .05$), reflecting compensatory visual strategies, but had a moderate negative impact on QE duration (Mean difference $= -161.45$ ms, $p = .0002$). Correlation analysis revealed a negative association between QE duration and swing accuracy under fatigue conditions ($r = -0.71$, $p < .05$ for mental fatigue and $r = -0.49$, $p < .05$ for physical fatigue), highlighting the role of sustained attention in motor performance. These findings demonstrate distinct cognitive and motor mechanisms underlying the effects of mental and physical fatigue, with implications for optimizing training and recovery strategies in sports.

## Introduction

Enhancing athletic performance is a primary objective for coaches, sports psychologists, and researchers, in the sports industry. In golf putting, a closed-loop aiming task uses continuous visual search to adjust and improve the putt's accuracy. Achieving success in golf putting depends on the ability of the golfer to accurately perceive

which permits unrestricted use, distribution, and reproduction in any medium, provided the original author and source are credited.

**Data availability statement:** All relevant data are within the paper and Supporting Information files.

**Funding:** The author(s) received no specific funding for this work.

**Competing interests:** The authors have declared that no competing interests exist.

the distance of a putt and the characteristics of the green, such as placing speed and the slope degree, and then translating that perception into an accurate action of striking the ball with the putter head [1,2]. The keynote of perceiving the position has a significant role in the enhanced performance of golf players, which Fatigue affects.

On the other hand, the Quiet Eye (QE) is a significant factor in successfully executing the extensively researched in different tasks. Longer last fixation, a QE to the putt, is a critical factor in successful performance [3,4]. QE is characterized as the final fixation, taking place before the athlete initiates an execution on a distinct spot or object for a minimum of 100 milliseconds. The onset of QE precedes the initiation of the action, with the endpoint signified by a deviation of 3 degrees from the targeted object [4]. While several theories have been proposed to explain this phenomenon, no single explanation has gained widespread acceptance or strong evidence to fully support it [5]. The information processing approach is frequently examined through empirical research [4,6]. Based on the information processing approach, athletes cognitively process relevant information during QE. This can involve preprogramming the main action or ensuring online performance during motor tasks [7]. The duration of QE increased when the task complexity was heightened, indicating a greater need for information processing. According to previous research, the duration of QE is affected by the complexity of the task, fatigue, expertise, the environment, and other factors related to the athlete [6,8,9]. An alternative theoretical concept, the Inhibition Hypothesis, suggests that the function of QE is to inhibit less or non-optimal motor reactions [10]. Furthermore, this hypothesis is supported by expert players, who possess a broader range of potential or alternative responses for specific situations, exhibit longer QE periods than novices. This indicates that QE may play a pivotal role in the optimal selection and suppression of less effective [10]. Vickers (1996) revealed that expert golfers exhibit prolonged fixations on the golf ball and fewer on the putting surface, while novice golfers allocate their fixations more evenly across different areas. Furthermore, maintaining focus on the ball throughout the swing and subsequently directing attention to the putting surface after contact has been linked to a higher likelihood of success [11]. Subsequently, Vickers (2007) found that less skilled players exhibited an irregular gaze pattern toward the ball, while skilled players sustained a prolonged fixation on the back of the ball during their swing, utilizing a more consistent scanning method from the ball to the golf hole [4]. According to the performance of expert golf players, an efficient QE in golf involves: (1) maintaining a long, unbroken focus on the back of the ball; (2) initiating this focus before the backswing; (3) sustaining fixation throughout the backswing, forward swing, and contact; and (4) pausing briefly after contact [4,12]. This approach suggests that longer final fixation on relevant visual-spatial details can reduce cortical resources usually associated with analytical processing and attending to irrelevant sensory signals [13]. Therefore, the period of QE seems to indicate the duration dedicated to processing visual information essential for motor control [11]. Different characteristics of the individual, the environment, and the task affect the duration of QEs. Fatigue is one of the factors that, like the complexity of the task, affects the duration of the QE period.

A significant factor that can decrease performance is fatigue, which can be categorized as either mental or physical [14]. Enoka et al. (2008) define muscle or physical fatigue as a temporary reduction in the ability to perform physical tasks and a decline in performance or the maximum force (muscle contraction velocity) that the muscles involved can generate [15]. This can be considered a symptom of blood occlusion, as the limited blood flow results in inadequate delivery of oxygen and nutrients, as well as insufficient removal of metabolic waste products, leading to an accumulation of lactate [16]. Those factors can significantly affect muscle coordination, which is reflected in player performance. Mental fatigue is described as a psychobiological condition where athletes display subjective, behavioral, and physiological changes resulting from extended periods or overly demanding cognitive tasks [14]. Warm and colleagues postulate that tasks demanding heightened alertness are stressful and entail substantial mental exertion [17]. Mental and physical fatigue shifts the focus of attention. It involves reduced attention toward the goal (goal-directed attention) and heightened focus on the irrelevant stimulus (stimulus-directed attention) [18,19]. Studies have examined how alterations in individual characteristics (fatigue) impact QE and visual processing [6]. Mathers and Grealy (2014) examined fatigue effects in elite golfers, finding that fatigue correlated with decreased putt success rates, altered golfers' scaling techniques, and extended QE duration. Additionally, increasing the demands of visual processing tasks extended the QE duration [20]. On the other hand, fatigue affects movement kinematics and coordination. Lees (2003) has indicated that fatigue results in greater performance variability, slower reaction times, and more errors across different sports performance metrics [21]. Specifically, Evans et al. (2008) revealed that fatigue from 40 minutes of golf putting practice caused notable changes in subsequent full golf swing patterns [22]. Moreover, fatigue negatively impacts performance accuracy in fine perceptuo-motor tasks [23,24], coordination during throwing [25], and can lead to poor timing, general lethargy, and either a loss of concentration or a shift in attentional focus detrimental to task demands [26]. While these findings suggest that fatigue may affect the mechanics of the golf putting stroke, other studies have indicated that expert performers can adapt their movement patterns to counteract fatigue and maintain spatial accuracy [27].

While earlier studies examined how fatigue affects visual search patterns and motor performance in sports [20,28], the majority of them have focused on either mental or physical fatigue independently, so disregarding a direct comparison of their effects on perceptual-motor performance. For sports such as basketball [29] and football [30], the impact of tiredness has been studied. This study addresses this gap by comparing the duration of mental and physical fatigue in the resting eye (QE), visual search behavior, and motor coordination during the execution of the golf swing. Few studies, however, have examined its impact on self-paced activities such as golf, where precision and sustained focus are critical. This study aims to comprehensively understand how fatigue affects cognitive-motor processes in fine motor skills by examining visual search patterns and motor coordination measures, namely radial error and swing accuracy. This study is one of the initial investigations to concurrently evaluate objective performance outcomes alongside subjective perceptions of effort (VAS and Borg scale), elucidating the link between perceived exertion and visual systems during fatigue. This study bolsters the expanding corpus of research on the cognition-action interaction paradigm, particularly in sports where performance relies on gaze behavior.

This empirical study emphasizes the necessity for an objective evidence to uncover the nature of information processing requirements during mental and physical fatigue. The effect of mental fatigue on QE duration have been well researched [31,32], but of the few studies that have investigated the comparative effects of mental, physical, and mental-physical effects on QE duration and motor coordination in golf swing. Thus, The first aim of the current study is to investigate the effects of mental and physical fatigue on gaze behavior, QED, and golf performance accuracy. The second aim of the current study is to investigate the correlation between QED and golf performance. . The second aim of the current study is to investigate the correlation between QED and golf performance. Based on past results and theoretical frameworks, we hypothesized that mental and physical tiredness would affect QE indices, motor coordination, and golf swing accuracy differently. Moreover, we hypothesized that under mental and physical tiredness, QED would

be adversely connected with radial error, thus highlighting the critical need for consistent attention in preserving motor performance.

## Materials and methods

### 1. Ethical considerations

The research protocol was developed following the Declaration of Helsinki and received approval from the Sport Sciences Research Institute of Iran with the permission ID IR.SSRC.REC.1401.120. All subjects submitted written informed consent before participation. The participants were fully informed about the study's purpose, procedures, and their right to withdraw at any time. The date range for participant recruitment was June 15, 2023, to September 10, 2023.

### 2. Participants

The sample size of 33 participants was determined using G*power (version 3.1.9), with the significance level set at.05, statistical power at 0.90, and effect size at 0.25 [33]. Repeated measures were analyzed using the ANOVA statistical test. The participants consisted of 33 undergraduate female students (mean age of $21.70 \pm 2.21$ years, average height of $165.73 \pm 6.09$ cm, and mean weight of $60.76 \pm 5.77$ kg), following a within-participant design. Inclusion criteria for the study were: (1) participants needed to be in good physical health without any joint or muscle injuries, and (2) ages should range from 18 to 30 years. Exclusion criteria included: (1) any history of neurological, musculoskeletal, or cardiovascular disorders that could interfere with motor performance or visual search behavior; (2) current physical injuries or pain affecting the upper or lower limbs that might impair golf swing execution; (3) uncorrected visual impairments or any condition affecting visual function; (4) use of medications that could alter cognitive function, attention, or physical performance; (5) failure to meet the physical fitness requirements necessary to complete the physical fatigue protocol.

### 3. Study design

In this study, 33 participants were initially recruited. However, 3 participants withdrew from the study, and 30 participants ultimately completed the protocol. The participants were randomly assigned to one of three groups: control, mental fatigue, and physical fatigue. All participants completed a golf performance in both the pre-test and post-test stages. Since the effects of fatigue are transient, the post-test session was conducted immediately following the mental or physical fatigue protocols.

The experiment was carried out in three phases for each participant: (1) a pre-fatigue baseline session, (2) a fatigue induction session (mental or physical), and (3) a post-fatigue session.

After obtaining consent, the participants were instructed on the proper method for golf swing. The familiarization session included an explanation of the measurement steps, correct way of performing a golf swing, and familiarization with the equipment. During the warm-up phase, all participants completed 15 golf swings emphasizing accuracy, without specific instructions on where to look. No measurements of accuracy were taken during this warm-up phase. Then, in the pre-test (15 trials, 5 blocks × 3 trials), the participants of all three groups performed the golf swing task from a distance of two meters [34]. The participants were asked to stop the ball as close to the hole. The distance from the edge of the ball to the center of the hole was measured as the radial error [34].

Golf performance (radial error), Quiet Eye (QE) duration, and gaze behavior were recorded during both the pre-fatigue and post-fatigue phases. In the fatigue session, participants completed the mental and physical Visual Analog Scale (VAS) assessments before and immediately after the fatigue protocol. After undergoing the mental or physical fatigue protocol, participants performed the golf swing task again. These assessments were then repeated during the post-fatigue phase.

The fatigue induction protocols were designed to replicate real-world fatigue commonly experienced in sports, drawing from previous research in the field. Mental fatigue was induced using the Stroop test, while physical fatigue was induced with a physical endurance protocol that included short sprints and golf-specific exercises.

## 4. Apparatus and task

**4.1. Eye-movement apparatus.** An eye-tracking system (Pupil Labs, Berlin, Germany) operating at 120 Hz captured gaze behavior during each condition. The system comprised two ocular cameras, each capturing at 120 Hz, alongside a singular environmental camera operating at 30 Hz. An extra smartphone camera (Apple iPhone 7 Plus, USA) was placed 5 meters to the right of each participant on the sagittal plane, capturing hand movements at 30 Hz as individuals executed straight-putting tasks. Before starting each set of trials, we calibrated the eye-tracking device and smartphone camera with a laser flash to guarantee alignment in the recorded video data. Subsequently, all videos were imported into Kinovea 0.8.15 video analysis software (https://www.kinovea.org/) to ensure the precise synchronization of gaze and motor data for subsequent analysis. The calibration of the eye-tracking technology device required players to adopt a standard golf stance and concentrate on five specific natural sights adjacent around the hole. Calibration was checked before each set of trials, and recalibration was carried out if necessary to ensure accuracy. The variables of fixation number, fixation duration, and QED were collected. The complete analysis of such variables using Pupil Player provided further detailed data on fixations, saccades, blinks, and pupil measures. Such a method allowed for an accurate analysis of gaze behavior aligned with motor coordination.

**4.2. Mental fatigue.** The Stroop test was used to induce mental fatigue. In this color-word task, participants pressed a key on the keyboard corresponding to one of four ink colors (yellow, red, blue, or green), disregarding the word's actual meaning and focusing solely on the ink color. This test induced cognitive interference, the Stroop effect, arising from the conflict between reading the word and recognizing its ink color [35]. In specific trials, participants were directed to disregard the original color-identification task and focus on the word's semantic meaning, thereby introducing an additional cognitive burden. A visual analog scale (VAS) questionnaire, validated with a content validity index of 72%, was utilized to evaluate mental fatigue [36].

**4.3. Physical fatigue.** We designed a 60-minute fatigue session based on the ideas of a golf performance coach and referencing studies on endurance and fatigue in golf training. We used similar studies to design the fatigue protocol [37,38]. Physical fatigue sessions include three sets of 12 exercises, each involving sprinting to various distances (6.70m, 3.88m, 1.98m). Each sprint lasted 20 seconds, which was followed by 20 seconds of rest. There was a two-minute rest between each set. In addition, replicated typical golf movement patterns, such as swing drills, rotational exercises, and dynamic movements aimed at endurance in the lower and upper body, were used. Participants were asked to exert maximum effort in all sets. Heart rate was continuously monitored using a Polar RS400 heart rate monitor to monitor physiological responses and ensure they maintained heart rates that approximated 90–95% of their maximal heart rate (HR_max), as recommended for simulating match fatigue in high-performance players [39–41] Before and after the session, participants completed a VAS questionnaire to assess subjective fatigue levels, as recommended in fatigue research, to capture immediate changes in perceived exertion and endurance capacity [42]. Before and after the physical fatigue session, participants filled out the VAS questionnaire.

**4.4. Perceived fatigue and exertion.** Perceived fatigue was assessed using a VAS ranging from 0 (no fatigue) to 10 (maximal fatigue and exhaustion), which captures momentary fatigue and is validated for physical activity [43]. Perceived exertion was assessed with the Borg scale which ranges from 6 (no exertion) to 20 (maximal exertion) and captures participants' perception of exercise intensity [44].

## 5. Data pre-processing

In golf putting, QE duration was defined as the final fixation directed within 1° of visual angle on the ball, lasting at least 100 ms, and occurring before the initiation of the backswing [4,12]. QE fixation data was defined using the Pupil Player

software. Trials were excluded from analysis if eye-tracking confidence fell below 0.8 for more than 1000 ms or if a QE fixation was absent. Only 2% of trials (mean = 1.49, SD = 0.97) were excluded, with the remaining trials (mean = 73.55, SD = 0.97) included for further analysis.

## 6. Statistical method

For each measure, QED, number, and duration of fixation, the three trials within each block for each subject were averaged. A 3 (groups) × 5 (blocks) was used for performance (i.e., RE), and a 3 (groups) × 3 (phases) with mixed model ANOVA was conducted to determine significant differences for QED (ms), fixation duration, and fixation number. If the main effect of groups and phases was significant, LSD post-hoc was used to investigate differences between blocks. Separate Pearson's correlation analysis between RE and QED

## Results

### Swing accuracy

The ANOVA revealed a significant main effect for blocks ($F_{(5, 17)}$ = 131.34, p = .001, $\eta p^2$ = .83), and the main effects of groups and interactions were not significant. The LSD post hoc comparisons showed significant differences between blocks 2, 3, 4, and 5 with the pre-test. Pre-test and block 2 (Mean difference = 1.83, SE = 0.44, p = .035), pre-test and block 3 (Mean difference = 2.18, SE = 0.40, p = .030), pre-test and block 4 (Mean difference = 2.86, SE = 0.38, p = .029), and pre-test and block 5 (Mean difference = 3.01, SE = 0.30, p = .020). There were also significant differences between Blocks 2 and 4 (Mean difference = 1.03, SE = 0.34, p = .040), 5 (Mean difference = 1.18, SE = 0.32, p = .05). Fig 1 shows the mean swing accuracy as a function of different conditions of fatigue.

### QED

The mixed ANOVA conducted on QED indicated a significant main effect for groups, $F_{(2, 27)}$=3.57, $P=.04$, $\eta_p^2$ = .21, and for phase $F_{(1, 27)}$=8.09, $P=.008$, $\eta_p^2$ = .23. In addition, the groups×phase interactions were significant $F_{(2, 27)}$= 22.94, $P=.001$, $\eta_p^2$ = .63. The Bonferroni's post hoc test showed significant differences between physical fatigue and control groups (Mean difference = −161.45, $P=.0002$), as well as between mental fatigue and control groups(Mean difference = −138.75, $P=.0009$). Fig 2 presents additional results, including the mean, standard deviation, and Bonferroni's post hoc test.

### Total fixation duration

The mixed ANOVA conducted on the fixation duration showed there was a significant main effect for the phase of evaluation, $F_{(1, 27)}$ = 7.22, $P=.01$, $\eta_p^2$ = .21. The comparison of mean revealed that fixation duration in the post-test was lower than pre-test. In addition, the main effect for groups was not significant, however, the interaction effect of groups and phase was significant ($F_{(2, 27)}$ = 8.86, $P=.01$, $\eta_p^2$ = .39). Post hoc analysis using Bonferroni's revealed the significant decrease in fixation duration in the mental fatigue group compared to the control group (mean difference = 67.50, $P=.001$), indicating that mental fatigue had a substantial detrimental effect on sustained fixation. Similarly, the physical fatigue group exhibited a significant reduction in fixation duration compared to the control group (mean difference = 57.1, $P=.0002$). Fig 3 presents additional results, including the mean, standard deviation, and Bonferroni's post hoc test.

### Total fixation numbers

The mixed-ANOVA analysis conducted on Total fixation numbers indicated that the main effect of phase ($F_{(1, 27)}$ = 5.68, $p=.02$, $\eta_p^2$ = .17) and the interaction between groups and phase ($F_{(2, 27)}$ = 51.12, $p=.01$, $\eta_p^2$ = .79) was significant. The main effect of groups was not significant (p > .05).

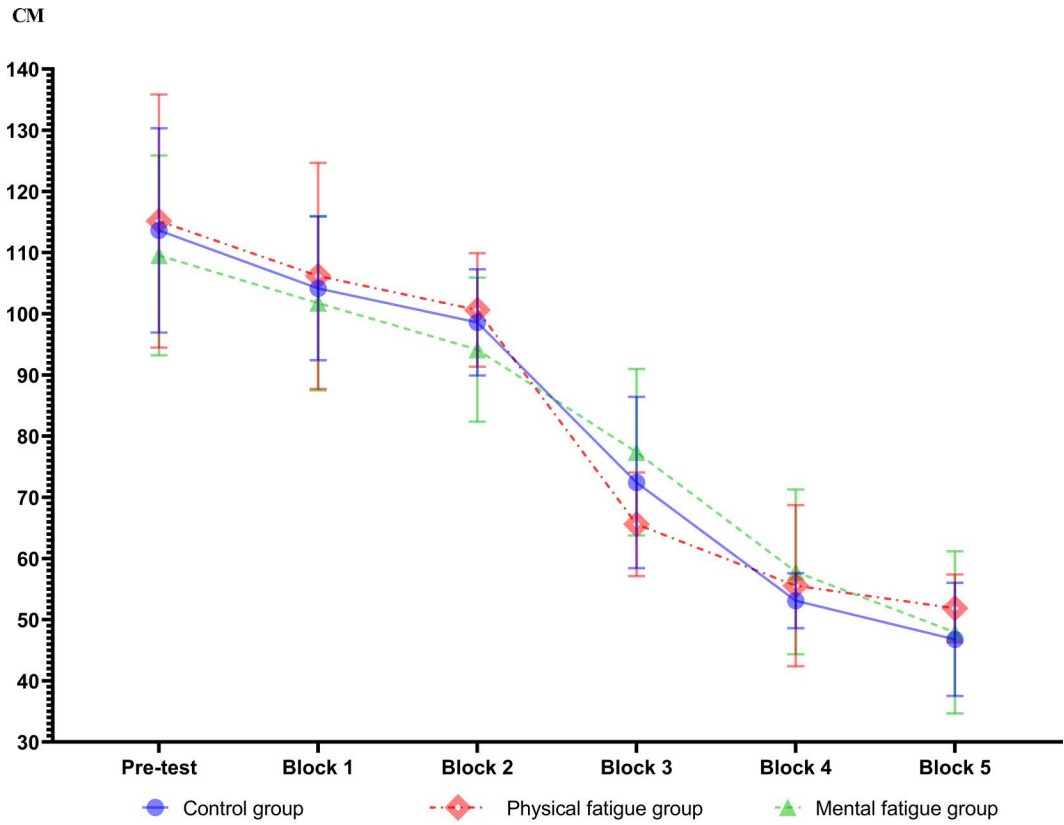

**Fig 1. Radial error (cm) as a function of different fatigue conditions.** Error bars indicate standard error.

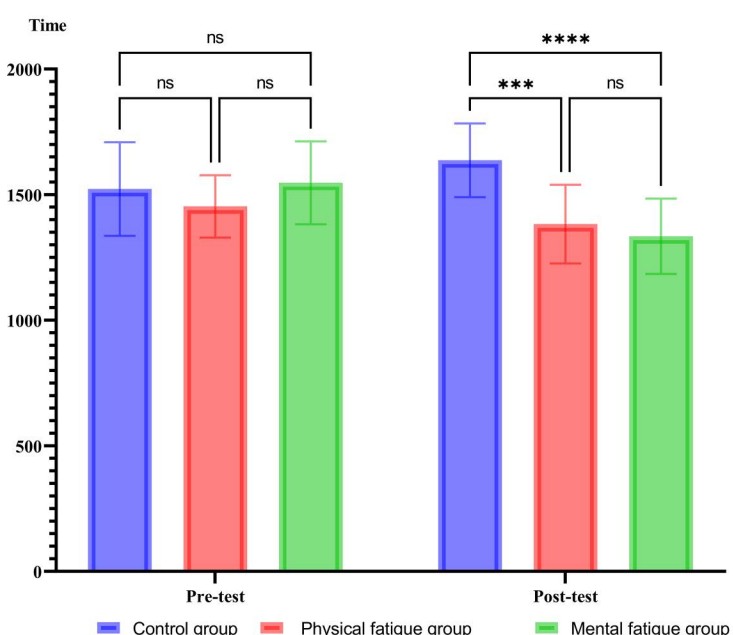

**Fig 2. Mean and standard deviation of QED, along with Bonferroni's post hoc test.** *Note: ***P ≤ .002 and **** P ≤ .009.*

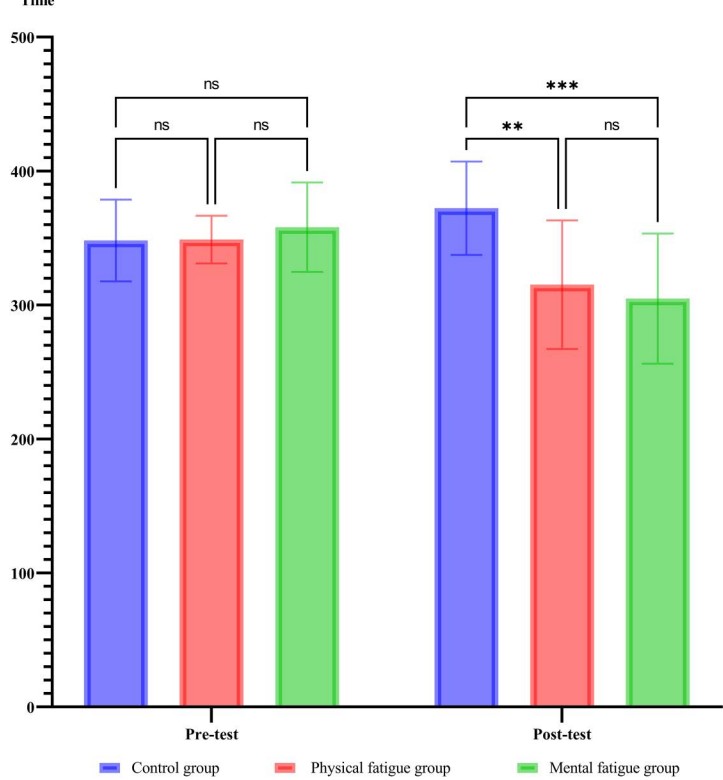

**Fig 3. Mean and standard deviation of Total fixation duration, along with Bonferroni's post hoc test.** *Note:* ** refer to *P ≤ .001* and *** refer to *P ≤ .0002.*

### Relation between QE and radial error

The Pearson correlation test showed that significant associations were revealed by the correlation analysis between radial error-mean swing accuracy and QE duration within various fatigue conditions. More precisely, the following were: a high negative value between mental fatigue and QE duration ($r = -0.71$, $p < .05$), suggesting reduced QE duration with increased mental fatigue. Regarding physical fatigue, the relation was moderate ($r = -0.49$, $p < .05$). This would mean that both mental and physical fatigue are negative towards QE duration and thus could affect swing accuracy.

### Discussion

This research is the first to investigate how physical and mental fatigue impact visual search, QE, and perceived effort in golf, aligning with our main hypothesis. The key findings indicated that mental fatigue leads to changes in visual search behavior, specifically in the number and duration of fixations, as well as QE. Consistent with earlier studies [30,45], subjective ratings of mental fatigue likely increased after completing the Stroop task but remained constant following the control. This difference is probably due to the greater mental effort demanded by the Stroop task. However, the findings of this study are not supported by later research; Fortes et al. (2022) noted that mental fatigue can shift attentional focus from goal-directed to stimulus-driven attention [29], which can lead to attentional blindness, where important cues are overlooked. For instance, mentally fatigued basketball players seemed to have fewer fixations on relevant cues, such as those on their opponents or teammates. Consequently, an effective search strategy that incorporates numerous fixations is necessary for players to

be aware of each other's positions, movements, and passing opportunities [29,46]. The reasons for this difference may be linked to the type and conditions of fatigue experienced. Contrary to our expectations, Smith et al. (2016) found that mental fatigue had little impact on visual search behavior in a soccer-related decision-making task [46]. Multiple factors could affect fixation duration due to fatigue, with longer fixation times possibly indicating a deficit in oculomotor disengagement. Bocca and Denise (2006) showed that fatigue had a greater impact on saccadic latency when the fixation was fixed on the screen compared to when the fixation was removed before the eye movement target appeared [47,48]. This pattern of findings is often understood as indicative of challenges in disengaging the oculomotor system from the fixation. Additionally, research indicates that damage to the parietal lobe results in difficulties disengaging attention [49], and it is well established that fatigue disrupts the function of the parietal lobe. Therefore, fatigue might affect the capacity to initiate eye movements [50]. Vincze and Jurchiş (2022) reported that QE duration decreased as fatigue increased. The result revealed that participants had a shorter QE in the last 20 balls compared to the initial 20 [28]. Additionally, athletes experiencing mental fatigue tend to struggle with accurately processing optimal decisions within the game context [29]. Furthermore, they require additional time to reach a more informed decision [14]. An exercise that demands maximum effort and leads to exhaustion engages the peripheral components of fatigue, which transmit sensory and nociceptive signals that become more pronounced as the intensity of the effort increases. These signals, which have already been impacted by mental fatigue and processed by the central nervous system, result in a heightened perception of exertion; as the exercise continues, individuals increasingly recognize the difficulty of the task. The longer the activity lasts, the greater the perception of effort becomes, while performance declines. This phenomenon is linked to the, which plays a crucial role in autonomic regulation during challenging cognitive and motor tasks and is also involved in effort-related decision-making during exercise [51].

Additionally, the results only partially supported our second hypothesis, as acute physical fatigue did not decrease fixation durations, though it did lead to an increased number of fixations. This finding aligns with previous research conducted in badminton (albeit not in a real-game context), which indicated that fatigue adversely affects operational processes, leading to a decline in the efficiency of gaze behavior [52]. The findings of Loiseau et al. (2021) indicated that acute physical fatigue led to an increased number of fixations on two specific points (the shuttlecock and the vacant area following an opponent's stroke), but did not alter fixation durations. Novice participants exhibited a disorganized visual strategy for gathering information due to their lack of experience. Overall, these findings indicate that fatigue negatively impacts gaze behavior. However, elite players maintained their fixation on a greater number of locations for the same duration during the high-intensity physiological workload protocol. Previous research has demonstrated that varying physiological workloads can alter gaze behavior [53]; however, in the study by Casanova et al. (2022), no significant differences between groups were found under high-workload conditions [54]. While distinctions were apparent during low-workload scenarios, they were less pronounced during high workloads, suggesting that players who excel in decision-making under low workloads may not necessarily perform better in high-workload situations. Additionally, the study by Casanova et al. (2022) found that shorter fixation durations occurred under low-workload conditions, which aligns with previously published findings [55,56]. Participants with higher tactical expertise may have been able to respond more quickly and selectively extract relevant information from their performance environment. Furthermore, recent evidence indicates that engaging in concurrent physical activities can influence cognitive task performance; however, there is limited agreement on whether these effects are beneficial or harmful [57]. Some studies have reported negative effects of concurrent physical load, leading to slower reaction times (RT) or decreased accuracy, referred to as cognitive cost [58]. Conversely, other studies have indicated positive effects, known as cognitive benefits [59], or shown mixed effects that depend on the type of cognitive tasks involved [60]. These inconsistent results complicate the understanding of how physical exertion affects cognition across various applied and translational contexts. While visual search under physical fatigue is generally quicker, it is also more susceptible to interference from distractors [61].

The results of this investigation provide coaches, athletes, and sports psychologists with valuable insights for the development of training and recuperation strategies for self-paced sports, including golf. The findings indicated that cognitive

fatigue has a detrimental impact on motor coordination and gaze behavior, as mental fatigue substantially impairs Quiet Eye (QE) duration and gaze behavior. This implies that cognitive recovery strategies, including mindfulness-based interventions and mental relaxation techniques, should be integrated into training programs in conjunction with physical recovery protocols to enhance performance under fatigue conditions. Furthermore, the evident negative correlation between radial error and QE duration underscores the significance of visual attention training in the preservation of motor performance. To improve the precision of golf swing execution, players may benefit from training that includes gaze behavior training techniques.

Participants reported higher mental fatigue, indicating that the Stroop color-word task was effective in inducing mental fatigue. However, the mental fatigue and physical fatigue groups did not show any significant differences in behavioral performance or gaze behavior. This is a limitation of the research. This mismatch may stem from the cognitive task's nature, which may not have been sufficiently demanding to elicit observable alterations in gaze behavior. The use of skill-based tasks with varying levels of difficulty in future studies may more accurately reflect real-game settings. Moreover, including electroencephalographic (EEG) recordings may provide more objective insights into the neurophysiological processes underlying mental tiredness and its impact on motor performance and gazing behavior.

## Conclusion

This study demonstrated that mental and physical fatigue negatively affects golf swing accuracy, Quiet Eye (QE) duration, and gaze behavior. The findings indicate that mental fatigue has a more detrimental impact on gaze behavior and motor coordination than physical fatigue. The significant negative correlation between QE duration and radial error highlights the critical role of gaze behavior in maintaining motor performance under fatigue conditions.

These findings provide valuable insights into the distinct cognitive and motor mechanisms underlying the effects of fatigue on perceptual-motor performance. Future research should explore neurophysiological measures to understand better how fatigue affects the interaction between attention, gaze behavior, and motor coordination in motor skills.

## Supporting information

**S1 File. Data file.**
(XLSX)

## Acknowledgments

We thank all participants who took part in this study. This research was conducted in the Alzahra University laboratory.

## Author contributions

**Conceptualization:** Maryam Khalaji, Parvaneh Shamsipour Dehkordi, Fatemehsadat Mousavian, Saeed Alboghebeish.

**Data curation:** Maryam Khalaji, Parvaneh Shamsipour Dehkordi, Fatemehsadat Mousavian, Saeed Alboghebeish.

**Formal analysis:** Maryam Khalaji, Parvaneh Shamsipour Dehkordi, Fatemehsadat Mousavian, Saeed Alboghebeish.

**Funding acquisition:** Parvaneh Shamsipour Dehkordi.

**Investigation:** Maryam Khalaji, Parvaneh Shamsipour Dehkordi, Fatemehsadat Mousavian.

**Methodology:** Maryam Khalaji, Parvaneh Shamsipour Dehkordi, Fatemehsadat Mousavian, Saeed Alboghebeish.

**Project administration:** Maryam Khalaji, Parvaneh Shamsipour Dehkordi.

**Resources:** Maryam Khalaji, Fatemehsadat Mousavian.

**Software:** Maryam Khalaji, Parvaneh Shamsipour Dehkordi, Fatemehsadat Mousavian.

**Supervision:** Maryam Khalaji, Parvaneh Shamsipour Dehkordi, Fatemehsadat Mousavian.

**Validation:** Maryam Khalaji, Parvaneh Shamsipour Dehkordi, Fatemehsadat Mousavian, Saeed Alboghebeish.

**Visualization:** Maryam Khalaji, Parvaneh Shamsipour Dehkordi, Fatemehsadat Mousavian, Saeed Alboghebeish.

**Writing – original draft:** Maryam Khalaji, Parvaneh Shamsipour Dehkordi, Fatemehsadat Mousavian, Saeed Alboghebeish.

**Writing – review & editing:** Maryam Khalaji, Parvaneh Shamsipour Dehkordi, Fatemehsadat Mousavian, Saeed Alboghebeish.

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
