## [Decision Letter · Decision Letter 0]

Dear Dr. Khalaji,

We look forward to receiving your revised manuscript.

Kind regards,

Rasool Abedanzadeh, Ph.D

Academic Editor

PLOS ONE

Journal Requirements:

3. Please note that your Data Availability Statement is currently missing [the repository name and/or the DOI/accession number of each dataset OR a direct link to access each database]. If your manuscript is accepted for publication, you will be asked to provide these details on a very short timeline. We therefore suggest that you provide this information now, though we will not hold up the peer review process if you are unable.

Reviewers' comments:

Reviewer's Responses to Questions

**Comments to the Author**

1. Is the manuscript technically sound, and do the data support the conclusions?

Reviewer #1: Yes

Reviewer #2: No

Reviewer #3: Yes

2. Has the statistical analysis been performed appropriately and rigorously?

Reviewer #1: Yes

Reviewer #2: No

Reviewer #3: Yes

3. Have the authors made all data underlying the findings in their manuscript fully available?

Reviewer #1: Yes

Reviewer #2: No

Reviewer #3: Yes

4. Is the manuscript presented in an intelligible fashion and written in standard English?

Reviewer #1: Yes

Reviewer #2: No

Reviewer #3: Yes

Reviewer #1: Interesting idea of this study, my recommendations are the following:

Abstract:

I recommend that abbreviations be first listed in parentheses after they have been mentioned descriptively.

Participants- I recommend mentioning the mean age and SD, gender and level of golf practice.

Results: I recommend revising, -Results are that..., possibly The recorded results reveal ..., or The results show that.....

Keyword_ I recommend adding a new word that refers to the sport referred to

Introduction- is well organized in terms of information, relevance and coherence. I recommend, at the end of the section, mentioning the objectives or hypotheses of this study.

Methods: I recommend introducing a new subsection called Study design where the typology of the study and other specific aspects should be mentioned.

I recommend mentioning whether the consent of the subjects to participate in the study was obtained.

I recommend that the subsections in the Methods section be numbered.

Lines 359-368 I recommend moving to the end of the Discussions section

Also, the practical implications of the study and future directions I recommend moving to the Discussion section.

In the conclusions section, only the main conclusions focused on the results should remain. I recommend reviewing the conclusions as they are practical implications.

Reviewer #2: The present study contains a number of shortcomings that are not the responsibility of the reviewer to list and analyse. The quality of the figures is poor and they are not legible, and it is not clear what is shown due to the authors not labelling the y-axis. The values given could be presented more clearly in a table, and the figures are unnecessary. The citations are a mixture of numbers and author+year format without following the journal format even approximately. Significant citations are absent, as evidenced by the example of line 128, which cites Abt et al., 2020, yet does not include the necessary author and year in the references.

Additionally, the exclusion criteria, delineated on line 133 as '(1) any unwillingness to cooperate throughout the study;', call into question the voluntariness of participant involvement, a fundamental requirement stipulated by the Declaration of Helsinki.

Reviewer #3: Thank you for the opportunity to review the manuscript with the title - Comparative Analysis of Mental and Physical Fatigue on Motor Coordination, Visual Search Patterns, Perceived Effort, and Performance Outcomes in Golf Swing Execution

The study is interesting and addresses a topical issue.

Recommendations for improving the content of the manuscript:

Abstract - we recommend including in the Results some relevant results identified in the study, currently only the textual interpretation of the results appears.

Introduction:

• To detail more specifically the novel aspects of the present study in relation to previous studies on the same topic.

• To formulate the hypothesis of the study.

Materials and Methods:

• Study design - To detail the periodization of the study: year, testing periods, stages of the study, etc.

Discussions:

- To add at the end of the Discussions - The practical implications of the study based on the relevant results.

- To delimit the limits of the study.

Conclusions

- We recommend restructuring the conclusions in relation to the main results identified in the study. It is not normal for bibliographical references to appear in the conclusions section. Conclusions refer to the relevant aspects of the present study and do not make connections to previous studies. Sentences where reference is made to previous studies should be moved to the Discussion section.

- Future research directions in accordance with the topic of the study should be mentioned

**Do you want your identity to be public for this peer review?** For information about this choice, including consent withdrawal, please see our Privacy Policy

Reviewer #1: **Yes: ** Badau Adela

Reviewer #2: No

Reviewer #3: No

---

## [Author Response · Author response to Decision Letter 1]

6 Mar 2025

Dear Rasool Abedanzadeh,

I appreciate you for your valuable comments. The submitted files meet all criteria that you mentioned in the letter. Thanks in advance.

Yours sincerely,

Maryam

---

## [Decision Letter · Decision Letter 1]

Comparative Analysis of Mental and Physical Fatigue on Motor Coordination, Visual Search Patterns, Perceived Effort, and Performance Outcomes in Golf Swing Execution

PONE-D-24-57374R1

Dear Dr. Khalaji,

We’re pleased to inform you that your manuscript has been judged scientifically suitable for publication and will be formally accepted for publication once it meets all outstanding technical requirements.

Kind regards,

Rasool Abedanzadeh, Ph.D

Academic Editor

PLOS ONE

Additional Editor Comments (optional):

Reviewers' comments:

Reviewer's Responses to Questions

**Comments to the Author**

Reviewer #1: All comments have been addressed

2. Is the manuscript technically sound, and do the data support the conclusions?

Reviewer #1: Yes

3. Has the statistical analysis been performed appropriately and rigorously?

Reviewer #1: Yes

4. Have the authors made all data underlying the findings in their manuscript fully available?

Reviewer #1: Yes

5. Is the manuscript presented in an intelligible fashion and written in standard English?

Reviewer #1: Yes

Reviewer #1: The authors have improved the article in accordance with all recommendations. All sections and subsections mentioned in this article, after revision, present complete and comprehensive aspects regarding the topic of the study. The article mentions the ethical aspects in accordance with the specifics of the study, both regarding the subjects and the study itself. The statistics applied are in accordance with the typology of the study.

**Do you want your identity to be public for this peer review?** For information about this choice, including consent withdrawal, please see our Privacy Policy

Reviewer #1: **Yes: ** Adela Badau

---

## [Editor Report · Acceptance letter]

PONE-D-24-57374R1

PLOS ONE

Dear Dr. Khalaji,

I'm pleased to inform you that your manuscript has been deemed suitable for publication in PLOS ONE. Congratulations! Your manuscript is now being handed over to our production team.

Kind regards,

on behalf of

Dr. Rasool Abedanzadeh

Academic Editor

PLOS ONE